# Effects of Red and White Ginseng Preparations on Electrical Activity of the Brain in Elderly Subjects: A Randomized, Double-Blind, Placebo-Controlled, Three-Armed Cross-Over Study

**DOI:** 10.3390/ph14030182

**Published:** 2021-02-25

**Authors:** Wilfried Dimpfel, Pierre-Antoine Mariage, Alexander G. Panossian

**Affiliations:** 1Departmrent of Pharmacology, Justus-Liebig-University Giessen, Germany c/o A 4164 Schwarzenberg am Böhmerwald, Panoramaweg 21, Schwarzenberg am Böhmerwald, Övre-Österrike, 4164 Österrike, Austria; 2NeuroCode AG, D-35578 Wetzlar, Germany; 3Botalys SA, 8 Quai des Usines, 7800 Ath, Belgium; pa.mariage@botalys.com; 4Department of Research & Development, Phytomed AB, Bofinkvagen 1, 31275 Vaxtorp, Sweden; ap@phytomed.se; 5Department of Science & Education, EuroPharmaUSA, Green Bay, WI 54311, USA

**Keywords:** elderly subjects, *Panax ginseng*, quantitative EEG, brain, discriminant analysis, cognition

## Abstract

Background: Recently, the superior efficacy of hydroponically cultivated red ginseng preparation HRG80^®^ compared to wild growing white ginseng (WG) in preventing stress-induced symptoms related to the daily work situation of healthy subjects was reported. The aim of this study was to compare the effects of HRG80^®^, WG, and placebo on the electrical activity in the brain of elderly human subjects during relaxation and mental challenges. Methods: Changes in the electroencephalogram (EEG) frequency ranges of 17 different brain regions were measured after single and repeated administration of HRG80^®^, WG, and placebo across a four-week randomized, double-blind, placebo-controlled three-armed cross-over trial. Results: Both red and white ginseng preparations had a strong impact on brain activity, with different effects on various brain regions depending on the mental load during relaxation and cognitive tasks associated with memory, attention, and mental performance. Both ginseng preparations exhibited significant effects on spectral powers compared to placebo, reflecting an activating action. The spectral changes in the quantitative EEG induced by HRG80^®^ indicated an improvement in mood as well as calming effects, evidenced by the modulation of β2 waves, representing changes in GABA-ergic neurotransmission. HRG80^®^ attenuated δ/θ powers during relaxation, suggesting the potential improvement of pathologically enhanced spectral power in aging. Conclusion: The results of this study suggest that both hydroponically cultivated red and wild growing white ginseng have similar beneficial effects on the cognitive functions of elderly subjects, as reflected by electric brain activity, but their modes of action on the brain are different.

## 1. Introduction

Ginseng Radix and Rhizoma (*Panax ginseng* Meyer) have been traditionally used in China, Korea, and Japan for thousands of years “to promote vitality and healthy aging, to enhance cognitive function, mental activity, general weakness, and enhance longevity” with long-term intake. It has been used primarily as a tonic to stimulate weak bodies and to maintain homeostasis, which is a contemporary term associated with adaptive stress syndrome, adaptability and adaptogen concepts [1,2,3,4,5,6,7]. According to the WHO [8], medicinal uses of ginseng include prophylactic and restorative treatment for enhancement of mental and physical performance, in cases of weakness, exhaustion, tiredness, and loss of concentration, and during convalescence. In Europe, most ginseng preparations are used as tonics in cases of tiredness, weakness, and decreased mental and physical capacity [9]. However, these and other international monographs [10,11,12] have been based on mixed data, without differentiating results from quite different preparations obtained from air-dried white ginseng (known in China as Renshen) or steamed at 100 °C to red ginseng (Hongshen) [13]. Chemical composition and pharmacological activity of white and red ginseng are obviously different [14,15,16,17,18,19].

Recently, the efficacy of white ginseng (WG) and hydroponically cultivated red ginseng (HRG80^®^) preparations in preventing stress-induced symptoms associated with the daily work situation of healthy subjects was reported [18,19]. HRG80^®^ was more effective compared to that with the WG and placebo regarding attention, memory, and perceived stress scores after single and repeated administrations for 5 and 12 days [18,19]. HRG80^®^ was also more active than WG in inducing excitatory neurotransmission of rat hippocampal slice preparations in an ex vivo model of long-term potentiation (LTP), reflecting time and spatial dependent memory [20]. It is well known that mental challenges are associated with significant changes in synaptic neurotransmission and differential electrical activity recorded at six electroencephalogram (EEG) brain frequencies (δ, θ, α1, α2, β1, and β2) in 17 different brain regions, which in turn correlate with various cognitive functions and mental diseases [21,22]. Furthermore, according to Dimpfel et al., 2014, high spectral δ and θ power (in general and specifically at frontotemporal regions of brain) is characteristic of mild cognitive impairment (MCI) [21]. The primary aim of present study was to detect possible differences in the effects of the same preparations (WG and HRG80^®^) on electrical activity of the brain of elderly human subjects in normal (relaxing) condition and during mental challenges (cognitive tasks), after repeated administration of test articles for four weeks.

## 2. Results

### 2.1. Baseline Data of Study Participants, Their Disposition and Treatment Compliance

Overall, 31 participants were assessed for eligibility; all met the inclusion criteria. Thirty patients were recruited in the study, of which all patients completed treatment and were included in the final analysis, Figure 1. One subject was withdrawn from the study (screening dropout) due to hypertension not related to trial medication. The treatment compliance calculated by counting of unused capsules was 98.97% (Placebo = 99.17%, HRG80^®^ = 98.42%, WG = 99.32%). 

Baseline demographic and efficacy outcome measures are shown in Table 1.

### 2.2. Efficacy of Treatment

#### 2.2.1. Efficacy Outcome Measures and Endpoints 

The primary efficacy outcomes were the responses of electric brain activity measured as spectral power in 17 different brain regions within six specially defined frequency ranges (i.e., δ, θ, α1, α2, β1, and β2) during relaxation as well as during the following psychometric tests of cognitive performance: concentration test for attention (d2 test), memory test (ME test), and calculation performance test (CPT).

The primary efficacy endpoint was achieved by comparison of the changes in neuronal electrical activity of the brain from baseline to the end of the treatment with WG, HRG80^®^, and placebo for four weeks.

#### 2.2.2. Quantitative EEG Results after Acute and Repetitive Dosing

Electric brain activity was recorded under several different conditions. The first recording was always in a relaxed state with open eyes. After this, three different mental challenges were presented during quantitative EEG recordings (Figure 2). 

The electric maps constructed after the intake of the placebo or one of the two ginseng preparations showed somewhat stronger differences for HRG80^®^ in comparison to placebo with respect to the frontotemporal brain areas under the eyes open recording condition and during the performance of the d2 test on the first day. 

Under recording conditions, the CPT and memory test after ingestion of either of the ginseng preparations induced similar patterns of change; however, they were only slightly different from that of placebo. The details are provided in Figure 2.

The EEG data were initially documented as absolute spectral power (µV^2^) for each electrode position (brain area) and each frequency range (δ–β2). The absolute power values from the baseline recording with respect to all recording conditions were then set to 100%. Drug-induced changes were documented as a pre-post intake comparison of the changes in percentage of these baseline values for every recording condition. When comparing the median values at the baseline (time 0) before all treatments, no major differences were detected. Thus, the starting values of the outcome measures were comparable. Drug-induced changes were documented as a comparison of the pre-post intake values as a percentage of the baseline values for each condition (Figure 3 and Figure 4).

On the last experimental day after the repetitive administration of HRG80^®^ for four weeks, statistically significant decreases in spectral α1 power in the frontal lobe F_z_, F_8_, and the temporal lobe T_6_, as well as α2 power in the central lobe C_3_, were observed 2 h after intake of HRG80^®^ during relaxation (Figure 3). With respect to the intake of placebo during relaxation, hardly any change was observed, whereas repetitive administration of WG during this period resulted in a statistically significant decrease in spectral α and β1 power in the parietal lobe at electrode position P_4_ (Figure 3).

As expected, quite different patterns were observed in the experiments when brain electrical activity was recorded during the different psychometric tests; namely, the d2 attention test, CPT (calculation test), and memory test. Under the recording conditions of the d2 test, WG changed spectral power only at the electrode positions C_3_, T_3_, and O_1_. HRG80^®^ induced statistically significant spectral changes in seven electrode positions; the details are documented in Figure 4.

A quite different response was observed in the CPT, where an increase in δ power was recorded in the frontal F3 lobe with decreases in θ power in the frontal F8 and α2 power in the temporal T5 location. Similarly, decreases in θ power in Cz and F8 as well as δ power in the Cz brain region were recorded during performance of the memory test. The EEG fingerprint response to WG treatment was different from that of HRG80^®^ in all three tests, although there were some similarities, e.g., only three of the 17 brain regions were significantly affected (decrease in α1 and α2 spectral powers) during the d2 test compared to the placebo. On the contrary, all the δ, θ, α1, α2, β1, and β2 spectral powers were significantly decreased in 12 of the 17 brain regions during the CTP, and the δ, θ, α2, β1, and β2 spectral powers were significantly decreased in 14 of the 17 brain regions during the memory test compared to the placebo. The details are provided in Figure 4.

#### 2.2.3. Efficacy of Red and White Ginseng Documented by Discriminant Analysis

All 102 parameters from the EEG recordings (17 electrode positions × 6 frequency ranges) during the eyes open condition were fed into the linear discriminant analysis containing data from several preparations earlier tested with identical methodology [23]. The results of both ginseng preparations projected in close vicinity of each other and were not well separated, also showing a similar color; altogether, this indicates that they have a similar action. The data following acute intake were similar to the effects of Adaptra Forte, another adaptogenic preparation [23]. The stimulating preparations Zembrin^®^ and memoLoges^®^ appear more toward the front, whereas HRG80^®^ and White Panax ginseng capsules appear toward the back. The details are shown in Figure 5.

### 2.3. Safety Evaluation

Thirty elderly subjects were administered HRG80^®^, WG, or placebo preparations at a daily dose of two capsules per day for four weeks. One participant dropped out during the screening, and there were no study dropouts. 

In total, four adverse events (AEs) were observed in three patients during the treatment with HRG80^®^, WG, and placebo capsules in phases A, B, and C of the study; for details, see Appendix A. These AEs were neither serious nor related to the study medication.

Overall, the treatment was well tolerated: all EEG measurements were within the normal range, and physical examinations did not reveal any deviation from normality. No organic disease emerged during the study; for details, see Appendix A.

## 3. Discussion

In the relaxation state, repeated administration of HRG80^®^ decreased the α1 and α2 spectral powers in six locations compared to the placebo, while WG decreased the α power in only one location and the β1 spectral power in two locations, Table 2.

Alpha1 spectral power is known to be associated with serotonin (5HT)-mediated neurotransmission. Thus, an increase in α1 power is positively correlated with relaxation and deep satisfaction (calming effect), whereas a decrease signalizes a higher activated state; meanwhile, α2 power is mainly associated with dopaminergic (DA)-mediated neurotransmission. A decrease in α2 is positively correlated with a stimulating effect, such as that induced by amphetamine, having an influence on mood and motivation that is possibly related to the pleasure–reward pathway. Attenuation of these waves must be interpreted as a potential positive action on mood. β1 power is associated with the modulation of glutamatergic (Glu) = mediated neurotransmission. A decrease in β1 power signalizes an activating effect (Table 2).

Since the electric activity of the frontal brain seems to be especially important for mental performance, analyzing this activity during treatment with drugs is essential. Even though both of the ginseng preparations seemed to generally act in a similar way during relaxation based on the results of the discriminant analysis, some differences were detected under the different challenging recording conditions and with respect to brain locations.

A major difference between the effects of HRG80^®^ and WG was observed during relaxation. Whereas HRG80^®^ reduced the spectral power in three frontal brain regions, WG did not affect any of the frontal locations under this recording condition (see Table 2). The effects of both ginseng preparations on brain activity depended considerably on the type of cognitive activity. During the d2 test, seven frontal brain locations were involved, in contrast to the effect of WG (no frontal regions involved). During the CPT, HRG80^®^ modulated three frontal regions, whereas WG had an influence on five frontal brain regions. However, HRG80^®^ affected more long waves, whereas WG mainly had an influence on high frequencies. During the memory test, six frontal regions were involved in response to both ginseng preparations.

These data are in accordance with the results of a recent study, where the beneficial effects of HRG80^®^ on stress were demonstrated in psychometric tests of healthy subjects [18]. These data are also in line with the observations in this study of increased δ/θ powers in elderly human subjects with MCI [21] since ginseng seems to reduce this pathological power. Table 2 shows that HRG80^®^, in general, also attenuated δ/θ powers in elderly human subjects during the memory test, but increased δ power at electrode position F3 during the CPT, indicating better performance during this test. The δ spectral power is known to be mainly associated with acetylcholine (Ach)-mediated neurotransmission; the activation of δ power in the presence of mental challenges is positively correlated with an improvement in learning, memory, and concentration, while the θ spectral power is mainly associated with the activation of norepinephrine (NE)-mediated neurotransmission related to attention. The β2 spectral power is mainly associated with the activation of GABA-mediated neurotransmission, which inhibits excessive excitation and is positively correlated with sedation and calmness: Strong sedative molecules such as diazepam induce large increases of β2 power. Therefore, ginseng seems to lead to more activation.

In contrast to red ginseng, WG inhibited one spectral frequency (α1) in only one electrode position (C3) during the d2 test for attention, where it was significantly less effective in comparison to HRG80^®^ (see also Mariage et al. [18]). During the present clinical trial, WG attenuated the electric power of all EEG waves during the memory tests (see Table 3) and was as effective as HRG80^®^ in the memory test (see also Mariage et al. [18]), where Ach-mediated neurotransmission plays a major role. WG also attenuated the power of all EEG waves during both the CPT and memory test, but in different regions of the brain.

In summary, both ginseng preparations acted in a similar and statistically significant manner in terms of the effects on electric brain activity of older subjects in comparison with placebo. However, some differences were also observed between red and white ginseng with respect to the involved brain regions, especially for the frontal lobe.

## 4. Materials and Methods

### 4.1. Participant Eligibility and Study Population

This clinical trial was performed at the contract research organization (CRO) NeuroCode AG (Wetzlar, Germany) with governing approval of an ethical committee at Landesärztekammer Hessen, Germany (approval date: 25 September 2019; protocol number: PP_0319_EuroPharma Ginseng HRG80^®^ Final V1 from 3 July 2019). This study was conducted in accordance with the current version of the Declaration of Helsinki (52nd WMA General Assembly, Edinburgh, Scotland, October 2000). The trial was conducted in agreement with the International Conference on Harmonization (ICH) guidelines on Good Clinical Practice (GCP). This trial was registered at ClinicalTrials.gov (accessed on 23 February 2021) (Identifier: 04167449; https://www.clinicaltrials.gov/ct2/show/NCT04167449 (accessed on 23 February 2021), Effects of Korean Red Ginseng Extract on Electrical Brain Activity in Elderly Subjects—Full Text View—ClinicalTrials.gov (accessed on 23 February 2021); last assessed on 26 February 2021). All participants gave written informed consent to participate in this study.

Thirty-one elderly volunteers of both genders aged 60–75 years were assessed for eligibility and enrolled in this study from October 2019 to March 2020. They were informed about the objectives, technical procedures, potential risks, and benefits of the study, and were asked to terminate taking dietary supplements that may have affected their cognitive functions at least two weeks prior to beginning the study. The inclusion and exclusion criteria were the same as in our previously reported study [23]—Pharmaceuticals 2020, 13, 45; doi:10.3390/ph13030045.

### 4.2. Study Design

This was a randomized, double-blind, placebo-controlled, three-arm crossover trial to compare the efficacy of red *P. ginseng* Meyer root preparation HRG80^®^ to traditionally harvested six-year-old white *P. ginseng* Meyer root (WG) and a placebo in elderly subjects. The effects of both ginseng preparations and placebo capsules, taken for four weeks, were studied (Figure 1 and Table 3).

### 4.3. Intervention and Comparator

The dietary supplement used in this trial was a 418 mg HRG80^®^ capsule (Botalys SA, 8 quai des usines, 7800 Ath, Belgium) containing 209 mg of hydroponically cultivated *P. ginseng* Meyer dry root powder (HRG80^®^) corresponding to 31.7 mg of total ginsenosides Rg1, Re, Rf, Rh1, Rg2, Rb1, Rc, Rb2, Rd, Rg6, Rh4, Rg3, PPT, Rk1, C(K), Rh2, Rh3, and PPD, including 25.9 mg of rare ginsenosides Rh1, Rg2, Rg6, Rh4, Rg3, PPT, Rk1, C(K), Rh2, Rh3, and PPD; 209 mg of inactive excipient rice flour.

Details of phytochemical analysis including HPLC profile are shown in our recent publication [18].

The second dietary supplement used in this trial was a generic 418 mg WG capsule containing 382 mg of wild growing white *P. ginseng* Meyer dry root powder (WG) corresponding to 9.8 mg of total ginsenosides Rg1, Re, Rf, Rh1, Rg2, Rb1, Rc, Rb2, Rd, Rg6, Rh4, Rg3, PPT, Rk1, C(K), Rh2, Rh3, and PPD, including 3.06 mg of rare ginsenosides Rh1, Rg2, Rg6, Rh4, Rg3, PPT, Rk1, C(K), Rh2, Rh3, and PPD, calculated as ginsenoside Rb1 (Botalys SA, Ath, Belgium).

The visually identical placebo capsules each contained 418 mg of rice flour. The label included the drug name, study code, and storage conditions. Reference samples were retained and stored at Botalys SA.

The participants received a package containing either WG, HRG80^®^, or placebo capsules. They were instructed to take two capsules in the morning with water for four weeks. After a four-week washout period, all participants started to undergo further test preparation for the next four-week period.

#### Randomization, Blinding, Allocation Concealment, and Evaluation of Compliance

Study preparations were randomly labeled by a qualified pharmacist (QP) and the random sequence of treatment codes was retained at the manufacturing site until the study was completed. Randomization, blinding, and allocation concealment were performed as previously reported [18], ensuring a double-blind design.

The time administration of investigational products was under the investigator’s control, assuring 100% compliance. It was calculated by counting the remaining capsules for each subject from the first to the last day of the study and verified by the study monitor at the end of the study. Unused capsules (two capsules in each package) were returned to the sponsor.

### 4.4. Study Procedures and Follow-up

During the visits, the patient was isolated in a quiet, darkened room, sitting relaxed in a comfortable chair. EEG data were recorded twice: at baseline before drug administration (eyes open for 6 min followed by eyes closed for 4 min, during the d2 test for 5 min, the ME test for 5 min, and the CPT for 5 min) and 120 min after drug administration (Figure 6). These test conditions were validated and standardized in our previous studies. Between measurements, the subjects relaxed in the leisure room. The experiments took place at the same time of day, starting from 07:00.

#### 4.4.1. Phase A, Screening and Training

On visit 1, the participants were checked for eligibility and informed about the details of the study, including restrictions—do not consume more than one cup of coffee a day (in the morning) during the study and do not take medicines or dietary supplements that may have potential effects on their cognitive functions. They signed written informed consent and passed a routine medical examination to be eligible for inclusion in the trial.

#### 4.4.2. Phase A, Treatment and Assessment

On day A (Figure 6), the participants completed all the cognitive functioning and stress tests in the morning (baseline). Then, the Principal Investigator (PI) randomly assigned the participants to a treatment. The participants orally took the phase A treatment and repeated all the tests 2 h after intake of the trial medication. The participants took two capsules per day for 28 consecutive days. On the day of treatment (day B in Figure 6), the participants performed all the cognitive functioning and stress tests. Phase A treatment was followed by a washout period of four weeks.

#### 4.4.3. Phase B, Treatment and Assessment

On the first day of phase B (day C in Figure 6), the participants completed all the cognitive functioning and stress tests in the morning prior to intake of the trial medication (baseline). Then, the participants orally took the phase B treatment and repeated all the tests 2 h after the intake of the trial medication. The participants took two capsules per day for 28 consecutive days. On the day of treatment (day D in Figure 6), the participants performed all the cognitive functioning and stress tests. Phase B treatment was followed by a washout period of four weeks.

#### 4.4.4. Phase C, Treatment and Assessment

On the first day of phase C (day E in Figure 6), the participants completed all of the cognitive functioning and stress tests in the morning prior to intake of the trial medication (baseline). Then, the participants orally took the phase C treatment and repeated all of the tests 2 h after the intake of the trial medication. The participants took two capsules per day for 28 consecutive days. On the day of the treatment (day F in Figure 1), the participants performed all of the cognitive functioning and stress tests. The study was then considered to be complete.

### 4.5. Efficacy and Safety Evaluation

#### Efficacy Outcome

The primary efficacy outcome measures were responses of electrical brain activity as spectral power in the 17 different brain regions within six specifically defined frequency ranges (i.e., δ, θ, α1, α2, β1, and β2) during the cognitive performance tests: d2 test, ME test, and CPT with financial rewards. EEG data were recorded bipolarly from 17 surface electrodes according to the International 10–20 system, with differences in Cz as the physical reference electrode to provide a common average reference (computer-aided topographical electroencephalometry: CATEEM^®^, MEWICON CATEEM-Tec GmbH, D94089 Neureichenau, Germany) using an electro-cap. For a detailed description of the procedure, please refer to reference [23].

The signals of all 99 electrode positions (17 real and 82 virtual) were subject to fast Fourier transformation (FFT) based on 4 s sweeps of data epochs (Hanning window). The data were analyzed from 1.25 to 35 Hz using CATEEM^®^ software. In this software, the resulting frequency spectra were divided into six frequency bands: δ (1.25–4.50 Hz), θ (4.75–6.75 Hz), α1 (7.00–9.50 Hz), α2 (9.75–12.50 Hz), β1 (12.75–18.50 Hz), and β2 (18.75–35.00 Hz) [24]. Frequency analysis was based on absolute spectral power values and calculated as source density [24,25].

### 4.6. Safety Outcomes

Safety outcome measures—the occurrence and severity of AEs—were recorded from the date of randomization until the end of the trial. All AEs observed by a subject or investigators were recorded during the trial in detail, e.g., description of the event, onset (date and time), resolution (date and time), maximum intensity, action taken, outcome, and causality.

### 4.7. Sample Size Considerations

Thirty patients were enrolled per treatment phase. The sample size was not determined during this pilot study. It was estimated based on the results of our previous studies carried out under a similar experimental design.

### 4.8. Statistical Analysis

Statistical analysis was done as previously reported [23]. Electrical activity of the brain was measured as absolute spectral power (µV2) in various experimental conditions.

Data from the first recording (baseline) were set as 100% and electrophysiological changes produced by the placebo or the WG and HRG80^®^ capsules are reported as percentage changes. WG and HRG80^®^ capsules versus the placebo were compared by evaluating the baseline recording of the last day in comparison to the recording 2 h after the intake. A nonparametric sign test was used for the comparison between the effects of the placebo, WG, and HRG80^®^. The *p*-values are provided for the statistics of the exploratory study.

Linear discriminant analysis according to Fischer was used for comparison with other drugs of herbal preparations, as previously reported [23].

## 5. Conclusions

Despite numerous studies conducted during the last few decades, ginseng is not yet recognized in Europe and the US as a medicinal product with well-established use. That is mainly due to the inconsistency of the results of clinical studies, where ginseng preparations were not identical, and therefore reproducible efficacy from one study to another has not been achieved. Reproducible efficacy and quality of wild ginseng cannot be achieved because its environmental conditions are not regulated. Standardized methods of cultivation and processing are required to assure reproducible quality and efficacy. The efficacy of cultivated preparations must be validated in clinical studies on humans, where the effects of cultivated ginseng (particularly hydroponically cultivated) on the central nervous system are compared with the effects of wild ginseng. To the best of our knowledge, the only publication on this topic is the publication of the results of our recent of study of red ginseng preparation HRG80^®^, where superior efficacy of hydroponically cultivated red ginseng compared to wild growing ginseng on the cognitive functions of healthy subjects was demonstrated (Mariage et al., 2020). However, no direct evidence on the brain activity of HRG80^®^ was available.

In this study, we, for the first time, demonstrate that:Red ginseng has an effect on the CNS in humansRed ginseng has an effect on the CNS in elderly subjects with mild cognitive impairmentsCultivated in standard conditions, red ginseng has an effect on the CNS of elderly subjects with mild cognitive impairmentsHydroponically cultivated in standard conditions, red ginseng has an effect on the CNS in elderly subjects with mild cognitive impairmentsThe overall effects of white and red ginseng on the electrical activity of the brain are different, suggesting different pharmacological activity of the red and white ginseng preparationsA treatment duration of 4 weeks seems to be sufficient to uncover the action of ginseng on the activity of the human brain

No side effects were observed, suggesting also longer treatment periods without unwanted actions.

CONSORT 2010 Checklist is available in Appendix B.

## Figures and Tables

**Figure 1 pharmaceuticals-14-00182-f001:**
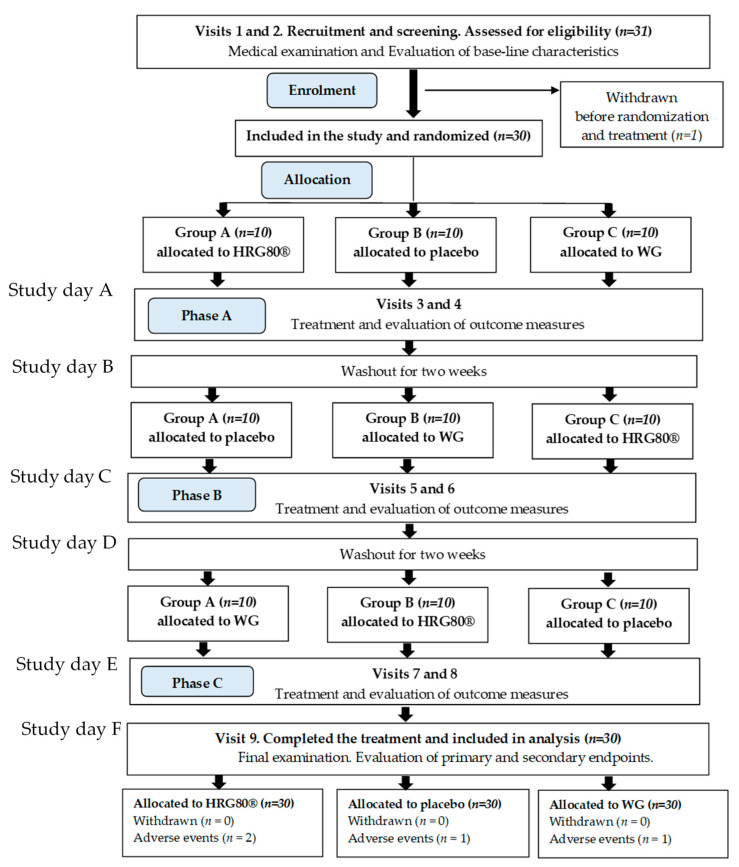
CONSORT flow chart of the disposition of participants into three arms on all steps of the study: screening, randomization, allocation, treatment, and data analysis. WG, white ginseng; HRG80^®^, hydroponically cultivated red ginseng.

**Figure 2 pharmaceuticals-14-00182-f002:**
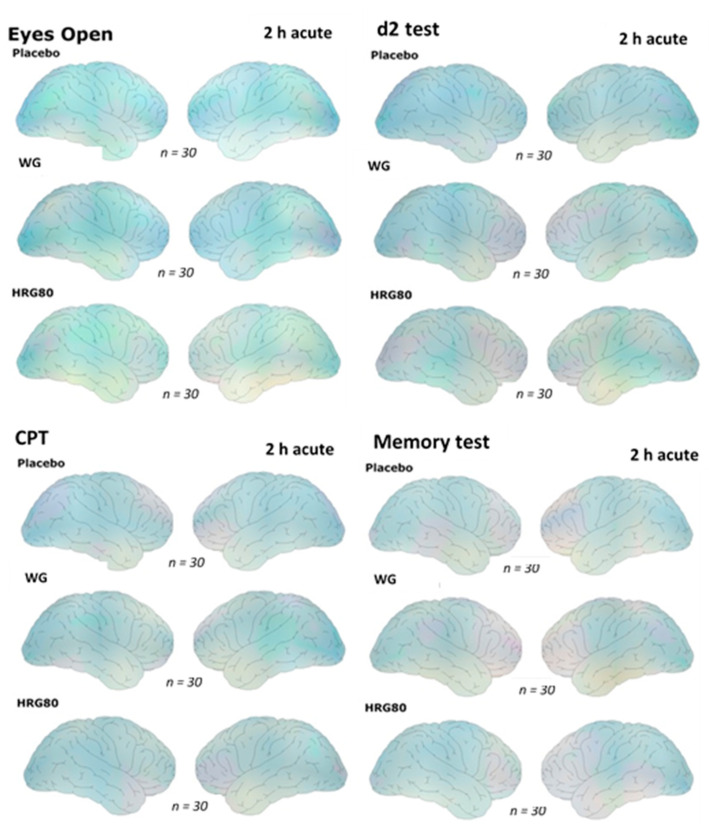
Changes in the electric brain maps 2 h after the ingestion of placebo, WG, or HRG80^®^ under different recording conditions on the first day of treatment. The right column in section represents the left hemisphere, while the left column represents the right hemisphere. Please note changes in the frontotemporal areas are more pronounced in the left hemisphere (frontal area is in the middle).

**Figure 3 pharmaceuticals-14-00182-f003:**
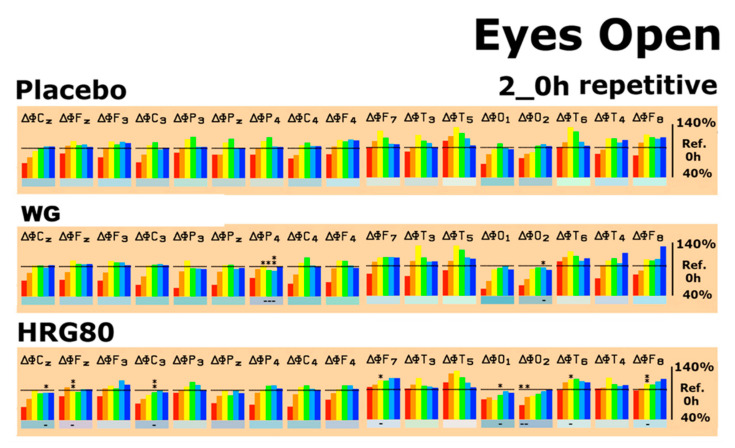
Spectral power differences under the eyes open recording condition in all brain regions, as represented by the electrode positions, on the last experimental day (repetitive) with the administration of placebo, WG, or HRG80^®^. C, central; P, parietal; O, occipital; F, frontal; T, temporal. Even numbers indicate a location in the right hemisphere, while odd numbers indicate the left hemisphere. Frequencies: red, δ; orange, θ; yellow, α1; green, α2; turquoise, β1; blue, β2. The spectral power (between 40% and 140% on the ordinate of the bar graph) was averaged over 6 min and plotted against the pre-drug value (values at 0 min set as 100%), thus reflecting the effect of the placebo, WG, or HRG80^®^. * *p* < 0.12, ** *p* < 0.05, and *** *p* < 0.01 (nonparametric sign test) between the placebo, WG, or HRG80^®^. The direction of change is marked by + or − underneath the relevant bar.

**Figure 4 pharmaceuticals-14-00182-f004:**
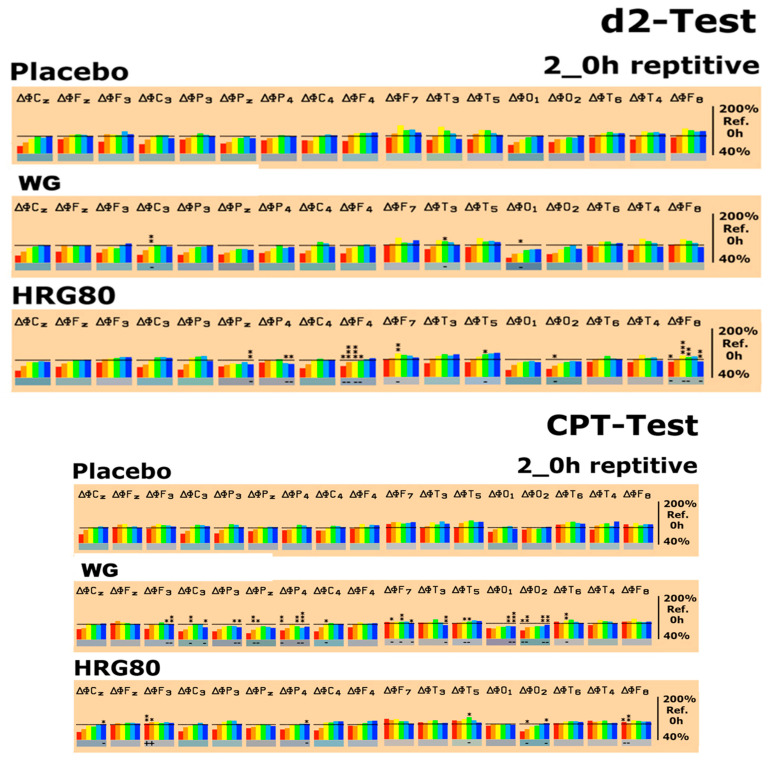
Spectral power differences in all brain regions, as represented by the electrode positions, under the recording conditions for the concentration test for attention (d2) test, the calculation performance test (CPT), and the memory test on the last experimental day with the administration of the placebo, WG, or HRG80^®^. C, central; P, parietal; O, occipital; F, frontal; T, temporal. Even numbers are located in the right hemisphere, and odd numbers in the left hemisphere. Frequencies: red, δ; orange, θ; yellow, α1; green, α2; turquoise, β1; blue, β2; h, hours. The spectral power (between 40% and 140% on the ordinate of the bar graph) was averaged over 5 min and plotted against the pre-drug value (at 0 min set to 100%), thus reflecting the effects of placebo, WG, or HRG80^®^. Statistical significance (sign test) between placebo, WG, or HRG80^®^ is indicated by stars. * *p* < 0.10, ** *p* < 0.05, and *** *p* < 0.01. The direction of change is marked by + or − underneath the bar graphic.

**Figure 5 pharmaceuticals-14-00182-f005:**
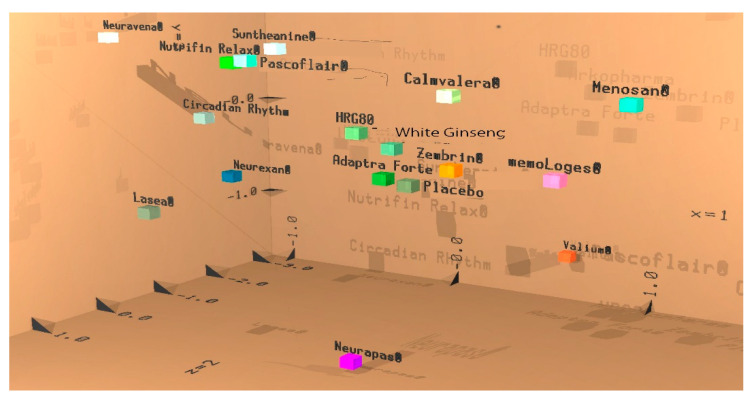
Results of the discriminant analysis. Acute effect on the first day of administration during the eyes open recording condition. White ginseng and HRG80^®^ are projected as near neighbors. Results from first three discriminant functions are displayed with respect to space (x, y, and z coordinates). Results from the next three functions are displayed as an additive color mixture. If the preparations are projected as rather close neighbors, they cannot be well discriminated from one another, which means that they have a similar effect or can be used for similar clinical indications.

**Figure 6 pharmaceuticals-14-00182-f006:**
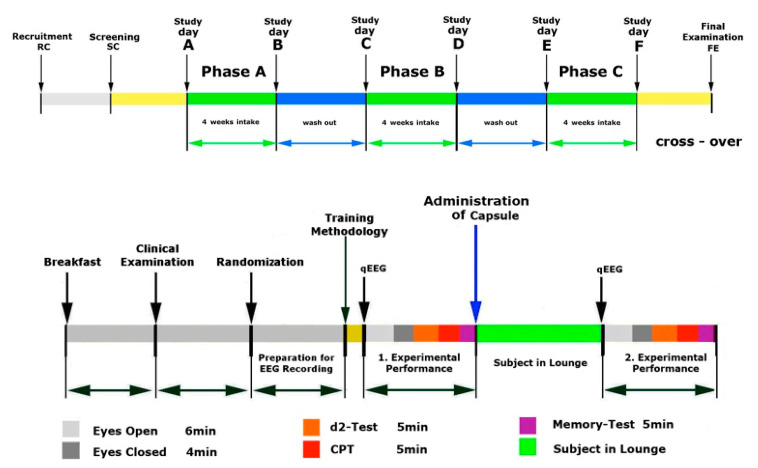
Design and timeline of the study days A, B, C, D, E, and F. Performance: eyes open (Eo), eyes closed (Ec), and different cognitive tests—attention test (d2 test), memory test (ME test), and concentration performance test (CPT).

**Table 1 pharmaceuticals-14-00182-t001:** Study population: baseline demographic characteristics.

Demographic Characteristics	*n*	Mean	SD
Age			
Total	30	65.63	3.68
Male	17	66.47	3.71
Female	13	64.54	3.48
Body height (cm)			
Total	30	1.74	0.10
Male	17	1.80	0.06
Female	13	1.65	0.06
Weight (kg)			
Total	30	79.80	14.49
Male	17	87.12	7.87
Female	13	70.23	15.81
Body Mass Index (BMI) (kg/cm^2^)			
Total	30	26.32	3.92
Male	17	26.85	2.44
Female	13	25.63	5.31

**Table 2 pharmaceuticals-14-00182-t002:** Statistically significant (*p* < 0.05) or conspicuous (*p* < 0.10) effects of HRG80^®^ and WG compared to placebo on spectral power (α1, α2, β1, β2, δ, and θ) under different recording conditions (relaxation and during the d2 test, CPT, and memory test) on the last experimental day. Brain regions are represented by the electrode positions. F, frontal; T, temporal; C, central; P, parietal; O, occipital. The direction of change is marked by **↑** or **↓** symbols.

Condition	Ginseng	δ	θ	α1	α2	β1	β2
Mediator		Ach	NE	5-HT	DA	Glu	GABA
Relaxation	HRG80^®^			↓Fz, F7, F8, T6	↓C3, O1		
	WG			P4	P4	↓P4, O2	
D2 test	HRG80^®^		↓F4	↓F4, F7, F8	↓F4, F8, T5		↓F8, Pz
	WG			↓C3, O1	T3		
CPT	HRG80^®^	↑F3, ↓F8	↓F8, O2		↓T5		↓Cz, P4, O2
	WG	↓P4	↓F7, Pz, O2	↓T5, T6, C3, C4	↓F7, T5, P4,	↓F3, P3, P4, O1, O2	↓F3, F7, T3, C3, O1, O2
Memory test	HRG80^®^	↓F7, F8, Cz, O1	↓F3, F8, Cz,	↓P4	↓Fz, F8		↓C3
	WG	↓T3, T6, O2	↓F7, T5, T6, C3, C4, P3, P4	↓F7, C3, C4, P4, O2	↓F3, F7, F8, T3, C3, P3, C4	↓F3, T3, P3,	↓T3, T5, C3, C4, P3

**Table 3 pharmaceuticals-14-00182-t003:** Challenges during the EEG recordings on the experimental days.

Challenges during qEEG Recordings on Study Days A, B, C, D, E and F at Baseline (0 h) and 120 min (2 h) after Intake of Two Capsules of HRG80^®^, WG, or the Placebo	
Eyes open (Eo)	6 min
Eyes closed (Ec)	4 min
Concentration test (d2 test)	5 min
Memory test (ME test)	5 min
Calculation performance test (CPT)	5 min
Total time excluding instructing of subjects	25 min

Note: h, hour.

## Data Availability

Data sharing not applicable due to privacy and force majeure reasons.

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
