# Peer review of "Effects of Red and White Ginseng Preparations on Electrical Activity of the Brain in Elderly Subjects: A Randomized, Double-Blind, Placebo-Controlled, Three-Armed Cross-Over Study"

_pharmaceuticals, 2021, doi:10.3390/ph14030182_

Round 1

Reviewer 1 Report

Dimpfel et al. investigate the effects of red and white Ginseng preparations on the electroencephalogramms (EEG) of elderly subjects in a clinical study conducted in a randomized, double-blind, placebo-controlled three-armed cross-over setting. Both preparations exhibited strong effects on the brain of the subjects as judged from the patterns of various brain waves. The authors conclude that both preparations have beneficial effects on cognitive functions of elderly subjects although their modes of action show differences.

The whole article is written in an excellent style. The method applied is described in a good way. The study is well performed and fulfills the criteria of the Consort checklist which is included at the end of the manuscript.

However, there are some issues which have to be changed and/or need further consideration.

General aspects

The heading of the articles mentions „Red and White Ginseng Extracts“, but actually the study medications consisted of „hydroponically cultivated P. ginseng Meyer dry root powder“ and „wild growing white dry root powder“ as parts of dietary supplements (see page 11). The use of the word „extract“ is misleading because no extraction process with solvents in a defined drug to extract ratio has taken place for the preparation of the dietary supplements (i.e. the study medication). Therefore the word „extract“ in the title should be replaced by „preparations“, „hebal preparations“ „root powders“ or „dietary supplements“.

The study was registered at ClinicalTrials.gov, identifier 04167449. However, the trial record on 04167449 differs from the study description in some points. In my point of view the deviations from the protocol should be mentioned and explained. E.g. the start of the study in the entry was Nov 1st, 2019, but in the manuscript (page 10/20) the start was already in October 2018 (and also prior to the approval date of the ethic committee in Sept 2019). Perhaps this is a only spelling mistake?

Further aspects

Page 1/20: the number 5 (EuropharmaUSA) is not attributed to one of the authors. Please check again!

Page 2/20: in the second paragraph („Recently ….“) the „of“ should be eliminated before „stress-induced symptoms“ („ … preventing stress-induced symptoms …“)

Page 3/20: In Fig. 1 I propose to include the terms „Study day A“, „Study day B“ etc. in the figure 1 consistent with Fig. 6

Page 9/20: Please explain the terms „slow waves“ and „fast frequencies“ for the readers who are not so familiar with quantitative EEG

Page 10/20: paragraph 4.1: Spelling mistake „Contract Research Organization“

Page 11/20: For the WG preparation the inactive ingredients should also be mentioned (like for HRG80)

Page 11/20: paragraph 4.3.1: Spelling mistake „… by the study monitor at the end…“

Page 14/20: The citation of the monograph on Ginseng radix in the European Pharmacopoeia is quite unusual, but not wrong. Currently, the version 10.0 is valid, although the original monograph (which was changed several times within the last years) might have been established in 2008. Therefore I propose to cancel the year 2008 and to cite as European Pharmacopoeia monograph 10.0/1523 (Ginseng radix) without a year. But this is up to you, and you may also leave the reference as it is!

Page 16/20: Table A1: Obviously, there is a spelling mistake regarding the description of the fourth case of an adverse event: in the left column the subject is called „no. 018“, whereas in the comment the same subject is called „no. 008“. Please check again!

In conclusion, this study is well performed with an excellent methodology and a well prepared manuscript on the effects of the two Ginseng preparations. In my point of view these findings will help to further improve the treatment of (mild) cognitive disfunctions in elderly people. Further research in this field will be stimulated based on the findings presented in this manuscript.

Reviewer 2 Report

I noticed that the approval of the study was obtained in 2019 and the recruitment of patients started in 2018. It would be well motivated.

Reference is made in the paper to figure 7 but which is not included.

In the conclusion, the results obtained from the research are not highlighted.

The paper does not bring novelty aspects being known the pharmacological properties of Ginseng Radix.

The results of the research contained in the article are useful to the manufacturer of nutritional supplements. The novelty of the research is not avoided.

A model of the patient's consent and the document certifying the approval of the ethics committee must be sent.

I recommend references from the last 10-15 years.

Round 2

Reviewer 2 Report

  I think it can be published.